# Current Concepts of Precancerous Lesions of Hepatocellular Carcinoma: Recent Progress in Diagnosis

**DOI:** 10.3390/diagnostics13071211

**Published:** 2023-03-23

**Authors:** Ziyue Liao, Cuiping Tang, Rui Luo, Xiling Gu, Jun Zhou, Jian Gao

**Affiliations:** 1Department of Gastroenterology and Hepatology, The Second Affiliated Hospital of Chongqing Medical University, No. 76 Linjiang Road, Yuzhong District, Chongqing 400010, China; 2020120766@stu.cqmu.edu.cn (Z.L.); tangcuiping117@stu.cqmu.edu.cn (C.T.); ruil@stu.cqmu.edu.cn (R.L.); 2Department of Pathology, The Second Affiliated Hospital of Chongqing Medical University, No. 76 Linjiang Road, Yuzhong District, Chongqing 400010, China; guxiling@cqmu.edu.cn; 3Department of Radiology, The Second Affiliated Hospital of Chongqing Medical University, No. 76 Linjiang Road, Yuzhong District, Chongqing 400010, China; zhoujun@hospital.cqmu.edu.cn

**Keywords:** hepatocellular carcinoma, precancerous conditions, pathology, diagnostic imaging, biomarkers

## Abstract

The most common cause of hepatocellular carcinoma (HCC) is chronic hepatitis and cirrhosis. It is proposed that precancerous lesions of HCC include all stages of the disease, from dysplastic foci (DF), and dysplastic nodule (DN), to early HCC (eHCC) and progressed HCC (pHCC), which is a complex multi-step process. Accurately identifying precancerous hepatocellular lesions can significantly impact the early detection and treatment of HCC. The changes in high-grade dysplastic nodules (HGDN) were similar to those seen in HCC, and the risk of malignant transformation significantly increased. Nevertheless, it is challenging to diagnose precancerous lesions of HCC. We integrated the literature and combined imaging, pathology, laboratory, and other relevant examinations to improve the accuracy of the diagnosis of precancerous lesions.

## 1. Introduction

It is estimated that primary liver cancer is the sixth most common cancer in the world and the third most common cause of cancer deaths [1]. Precancerous lesions are characterized by a multistep process ranging from dysplastic nodule (DN) to early HCC and finally to advanced HCC [2]. Cirrhosis was considered a precancerous condition associated with a high probability of developing HCC [3]. Although liver transplantation, ablation, and surgical resection have achieved good 5-year survival rates in the treatment of early-stage HCC, only about 20% of patients with HCC are diagnosed in the very early and early stages. The overall survival rate of patients with advanced liver cancer treated with transcatheter arterial chemoembolization (TACE), radioembolization, and systemic therapy is lower than that of patients with early-stage liver cancer treated with radical treatment [4]. Hepatitis B, cirrhosis, and liver cancer comprise the “trilogy pattern” of how HCC develops. As a result of cirrhosis, DN is more likely to develop malignant growths and exceptionally high-grade dysplastic nodules (HGDN). In order to improve patient survival, the focus and challenge of HCC diagnosis and treatment has shifted towards improving the detection of precancerous lesions and the early diagnosis of patients.

This article aims to comprehensively review the current diagnosis of precancerous lesions based on pathological, imaging, and laboratory examinations. In addition, the mechanisms of precancerous transformation and the clinical diagnosis of precancerous lesions were also discussed.

## 2. Concept and Classification of Precancerous Lesions of HCC

Approximately 80% of HCC is the result of regenerative nodules (RN) underlying cirrhosis (Figure 1A), and the main features of progression from cirrhosis to HCC are the occurrence of atypical hyperplastic foci (DF) and DN [5]. The diameter of DF is less than 1mm, often with prominent fibrosis. It may be categorized as large cell change (LCC) or small cell change (SCC) [6,7]. SCC may represent a more advanced disease compared to LCC. In contrast, according to the cellular and structural atypia stage, DNs greater than 1 mm in diameter and frequently occurring in cirrhosis are divided into low-grade dysplastic nodules (LGDN) and HGDN [8,9]. The annual liver cancer conversion rate was 10% in LGDN patients and 20% in HGDN patients. This comparison suggests that DNs, especially HGDN, may represent precancerous lesions [10].

### 2.1. Dysplastic Foci

SCC hepatocytes were reduced in size, homogeneous in appearance, and had an increased nucleoplasmic ratio, mild nuclear pleomorphism, pigmentation, cytoplasmic thickening, clear cytoplasmic borders, increased cytoplasmic basophils, and nuclear crowding [11].

According to the original definition, LCC occurs when the nucleus and cytoplasm are enlarged (so that the nucleoplasm ratio remains constant) and when there is nuclear pleomorphism, dense staining, and multinucleation [6,12]. Multiple heterogeneous lesions can accompany LCC [13]. Insufficient evidence suggests that the lesions are precancerous [14].

### 2.2. Dysplastic Nodule

The progression of HCC is multistep, alternating between LGDN and HGDN, early HCC (eHCC), progressed HCC (pHCC), nodular-in-nodular HCC, and finally, moderately differentiated HCC [15]. The histological features of DN are progressive nuclear crowding, structural disorganization, and a variable number of unpaired arterioles or capillaries [16].

LGDNs (Figure 1B) have a similar histopathological appearance to regenerative nodules since they have preserved liver structure, minimal cellular atypia, a normal vascular profile, normal hepatocyte function, and a normal Kupffer cell density [5]. The nucleo-cytoplasmic ratio was normal or slightly elevated. In the portal structures, pseudo-glandular ducts and isolated arterioles did not exist. There was an accumulation of iron or copper in the liver and uniform steatosis without fatty changes, which is a hallmark of clonal populations of cells. LCC may also be detected at the same time [17]. Despite its low malignant potential, it rarely causes HCC due to its slow and uncommon disease progression [16].

For HGDN (Figure 1C), the condition consists of structural distortion and more severe cellular atypia, including sinusoidal capillarization and an increase in unpaired arteries [18]. Kupffer cells were either increased, normal, or decreased in density [19,20,21]. The hepatocyte plate was 2 to 3 cells wide, with a high nucleo-cytoplasmic ratio, overstained nuclei, pseudo-glandular formation, and basophilic cytoplasm. The atypicality of the nodules is at least moderate but does not reach a diagnosis of malignancy. No invasion occurred in the mesenchyme or portal system [15]. There may be isolated arterioles with dilated growth associated with isolated arterioles. The combination of HGDN and SCC has been reported as a late event in the progression of DN, suggesting that the tumor is precancerous [22]. From a histopathological standpoint, it is challenging to identify HGDN from eHCC (Figure 1D).

Hyperplastic nodules have typical sarcoid recognition features, LGDN may evolve into HGDN, and features of malignant evolution include arterial recruitment (usually in HGDN) as well as stromal infiltration (eHCC), venous infiltration and metastatic growth (pHCC) [23]. Regarding precancerous lesions, the risk of malignancy in HGDN is four times higher than that in LGDN [16,24].

## 3. Molecular and Genetic Mechanism of Hepatocarcinogenesis

The molecular process of HCC carcinogenesis results from the accumulation of genetic and epigenetic alterations. There is a progressive increase in somatic copy number alterations (SCNAs), chromosomal instability, and telomerase reverse transcriptase (TERT) promoter mutations associated with hepatic dysplasia and HCC on the chromosomal arms in patients suffering from dysplasia of the liver and HCC [25]. Telomerase expression was found to be reactivated in tumors of 90% of HCC patients [26]. The activation of β-catenin has been observed in 20–35% of HCC cases [27]. One of the most frequently mutated genes in primary HCC is CTNNB1, which encodes β-catenin. However, the low prevalence of these mutations in DN suggests that they play a more critical role in tumor progression in cirrhosis than in the onset of carcinogenesis [28,29]. Chromosomal alterations, such as copy number variants, have been described in precancerous lesions, mainly including increases or deletions of chromosome 8 arms (8p or 8q) and increases in 1q [30]. An analysis of the prevalence of SCNAs at the arm level showed that 1q+ and 8q+ were common in HCC, uncommon in DN, and least common in cirrhosis [31,32].

## 4. Diagnosis of Precancerous Lesions of HCC

Precancerous lesions of HCC usually develop based on liver cirrhosis. They are typically less than 2 cm in diameter and do not exhibit any obvious clinical symptoms. Therefore, early detection is difficult. Although histopathology is the gold standard for diagnosing precancerous lesions, the widespread use of invasive tests is impractical. In clinical practice, precancerous lesions are rarely detected and are usually diagnosed by a combination of pathology, immunohistochemistry, and imaging examinations. Detecting precancerous lesions at an early stage has the potential to prevent and control the development of HCC.

### 4.1. Immunohistochemistry

#### 4.1.1. Glypican-3 (GPC3)

GPC3 is a carcinoembryonic antigen, a member of the phosphatidylinositol proteoglycan family of membrane-bound acetylated heparan sulfate proteoglycans, expressed in fetal liver and progenitor cells as well as in many HCC cases and other tissues [33]. According to a study by Coral et al., GPC3 has a sensitivity of approximately 65% and a specificity of 96% for diagnosing HCC [3]. Immunohistochemistry and a real-time reverse transcription-polymerase chain reaction showed that the expression levels of GPC3 were much greater in small HCC than in cirrhosis and other categories of minor lesions. As a result, in most cases, a dramatic increase in GPC3 expression occurs when precancerous lesions are transformed into small HCC [34]. Research has indicated that the expression rate of GPC3 in HGDN is higher than that in LGDN [35]. Based on the evidence presented in the current study, GPC3-positive DN, particularly GPC3-positive HGDN, can be used as a marker to differentiate HGDN from eHCC [36].

#### 4.1.2. Glutamine Synthetase (GS)

Overexpression of GS can result from mutations in β-catenin or the activation of this pathway [37]. The main energy source of tumor cells is glutamine, which is synthesized by GS [38,39]. There is substantial evidence that GS is expressed in 13–70% of eHCC and 10–15% of HGDN. The expression of GS in HGDN is typically focal [40,41]. Nevertheless, HSP, GS, and GPC3 failed to separate well-differentiated HCC from HGDN [40,42].

#### 4.1.3. Heterogeneous Nuclear Ribonucleoprotein A3 (HNRNPA3)

HNRNPA3 is an intranuclear RNA-binding protein that forms a complex with pre-mRNA [43]. Although there is a lack of information on the exact relationship between HNRNPA3 and tumorigenesis, members of the HNRNP protein family are closely associated with tumor regulation [44,45]. Xinlu Ren et al. found that the expression of HNRNPA3 increased as the liver tissue evolved from non-neoplastic to HCC. Furthermore, the expression of HNRNPA3 could serve as a useful marker for distinguishing HGDN from eHCC and appears to be more effective than other available biomarkers. In this way, HNRNPA3 can be used as a potential diagnostic marker for identifying specific stages in the development of HCC. Overexpression of this enzyme may also indicate a poor prognosis [46].

#### 4.1.4. miRNA-96-5p/3p

Growing evidence suggests that certain miRNAs can function as tumor suppressors or oncogenes by directly or indirectly regulating the expression of essential proteins involved in the development of hepatocarcinogenesis [47]. A significant increase in miRNA-96-5p levels was observed in normal liver tissue, cirrhosis, and HCC. Besides mature miRNA-96-5p, guest chain miRNA-96-3p was also involved in hepatocarcinogenesis, although these miRNAs displayed opposite expression patterns in multistep hepatocarcinogenesis. As a result of the negative expression of miRNA-96-3p, the sensitivity and specificity were 88.2% and 84.2%, respectively. When miRNA-96-3p was combined with GPC3 immunostaining, the sensitivity and specificity were 67.7% and 100%, respectively. Several studies have suggested that the miRNA-96-3p expression level is a useful marker for distinguishing between HGDN and HCC [48,49].

In patients with chronic hepatitis B, miR-122 and let-7b levels may be useful biomarkers to distinguish early HCC from DN [43]. However, this finding requires further verification through a large sample study conducted by several research teams.

#### 4.1.5. Heat Shock Protein 70 (HSP70)

In addition to being a member of the heat shock protein family, HSP70 plays a role in tumorigenesis, cell cycle progression, and apoptosis [42,50]. HSP70 is immunoreactive in a large proportion of HCC, including early and well-differentiated cancers, but not in non-malignant nodules [40]. Among the 12,600 genes studied by Chuma et al., the expression of HSP70 in eHCC was significantly higher than that in DN, and it was significantly higher in pHCC compared with eHCC. The presence of HSP70 in the liver may serve as a sensitive marker for differentiating eHCC from precancerous lesions or non-neoplastic liver tissue [51].

#### 4.1.6. Organic Anion Transporter Polypeptide (OATP)

Several experimental studies have demonstrated that the uptake and excretion of gadoxetic acid by hepatocytes are mediated by OATP1B3/1B1 and multidrug resistance-associated protein 2, respectively [52,53]. Kitao et al. found that OATP1B3 expression levels in LGDN were identical to those in the surrounding liver. On the other hand, 30% of HGDN and 75% of eHCC exhibited reduced OATP1B3 expression. There was a reduction in or absence of OATP1B3 expression in all poorly differentiated HCCs [54]. OATP1B3 expression was significantly decreased with the degree of tumor differentiation, making it a useful indicator of HCC, particularly at an early stage [55].

#### 4.1.7. CD34/CD105

As shown by CD34 immunostaining, sinusoidal capillarization was uncommon in cirrhosis and gradually increased from LGDN to HGDN, with the highest levels in HCC [56]. CD105 expression, however, gradually decreased throughout this process [57]. In an immunohistochemical study, CD34 and CD105 were used to determine microvascular density, which can assist in the differentiation of DN from HCC [58].

#### 4.1.8. Telomerase Reverse Transcriptase (TERT)

A small percentage of DN (6% of LGDN and 19% of HGDN) have TERT promoter mutations, and the incidence rises with the degree of heterogeneous hyperplasia [59,60]. The mutation of the TERT promoter in cirrhosis leads to the transformation of precancerous lesions into cancerous lesions [61].

#### 4.1.9. Combination

Multiple malignancy markers are combined to improve overall accuracy. The most representative HSP70/GPC3/GS combination is the “all-positive” phenotype (+/+/+), present in 43.75% of eHCC-G1 and never present in HGDN. In contrast, the opposite “all negative” phenotype (−/−/−) is characteristic of 72.73% of HGDN cases, although it also appears very sporadically in eHCC-G1 [40]. In clinical practice, the leukemia inhibitory factor receptor (LIFR) + CD34 combination had a useful differential diagnostic model for well-differentiated HCC and HGDN with a sensitivity and specificity of 93.5% and 90.5%, respectively [62].

### 4.2. Imaging Examination

Intra-nodular vascular alterations occur during hepatocarcinogenesis, with a progressive reduction in the portal blood supply and a concomitant rise in the arterial flow delivered by the so-called “unpaired” arteries. The arterial phase hyperenhancement (APHE) and wash-out of the contrast medium in the portal and/or delayed phases are the classic radiologic characteristics of HCC, which are caused by vascular derangement [63]. The typical radiographic characteristics of “wash-in” and “wash-out” are adequate for recognizing HCC in nodules larger than 10 mm, and a biopsy should only be considered for nodules with atypical imaging characteristics [64,65].

The histological characteristics of hepatic nodules are correlated with imaging findings [66]. OATPs can express the activity of hepatocytes, which is significantly reduced in hepatoma cells compared with benign lesions, leading to the hypointensity of the hepatobiliary phase [54,67]. In pHCC, the increase in new arteries leads to a wash-out manifestation in the arterial phase, which can also be distinguished from the low blood supply in HGDN and eHCC [68]. At the same time, the content of lipid and iron in the nodules is inversely proportional to the degree of malignancy, which makes lipid and iron deposition in the lesions a typical feature of precancerous lesions and large eHCC. It is manifested as hyperintensity on T1-weighted images and hyperintensity on T2-weighted and T2*-weighted images, respectively [67,69,70,71].

#### 4.2.1. Ultrasound (US)

Several studies have demonstrated that regular screening for liver cancer by abdominal ultrasound every six months and receiving curative treatment can significantly improve survival rates [72,73,74]. Color Doppler ultrasonography is mainly used as a screening tool, which is capable of detecting intrahepatic nodules based on size, intensity, and uniformity, but is not very useful in the diagnosis of eHCC.

In CEUS, microbubbles of a contrast agent, of a diameter of approximately 3–5 mm, sufficient to pass through the pulmonary circulation, were administered by intravenous injection. Within approximately five minutes after injection, it should show vascular enhancement and slowly diffuse into the bloodstream [75]. Consequently, repeat injections of contrast can be performed in approximately five-minute intervals. In the process of developing malignant nodules from DN, mismatched arteries of new angiogenesis gradually replace the intratumoral portal vein bundles. Thus, overt HCC is primarily perfused by the hepatic artery, and branches of the portal vein primarily perfuse normal liver parenchyma and precancerous nodules [8]. The delayed phase is characterized by predominantly hyperechoic or isoechoic lesions relative to adjacent parenchymal liver tissue, while the acoustic properties of malignant lesions are predominantly hypoechoic. Additionally, arterialization of the blood supply may account for the excessive enhancement of malignant nodules during the arterial phase and the ultrasound washout phenomenon during the portal and sinusoidal phases (a negative enhancement of the liver lesion in relation to the liver in the late phase) [76]. The irrigation time was related to the pathological differentiation of the nodules; well-differentiated nodules tend to wash out late or not at all, while poorly differentiated nodules tend to wash out rapidly [77]. Typically, LGDN exhibits a “slow in, wait out” enhancement pattern, with most showing equal enhancement. It has a variety of manifestations, ranging from persistent hypointense signals, early hypointense signals during the arterial phase, isointense signals in the arterial phase, hyperintense signals in the arterial phases, and mild hypointense signals in the portal venous and/or delayed phases, which is similar to the findings of contrast-enhanced ultrasonography in eHCC [78]. A pattern of fast-in-fast-out is observed for HCC, whereas a pattern of slow-in-slow-out is observed for DN [79]. The use of computed tomography (CT) or magnetic resonance imaging (MRI) may fail to detect arterial phase hyperenhancement (APHE) due to missed arterial phases, and contrast-enhanced ultrasound (CEUS) is superior to CT and MRI in the detection of nodal vascular hyperfunction [80,81]. CEUS has demonstrated significant improvements in the accuracy of US in identifying focal liver lesions due to the ability to distinguish between benign and malignant lesions based on vascular patterns in the arterial, portal, and sinusoidal phases [82,83]. Imaging fusion, which combines CEUS and MRI, is a promising technique for improving the detection, accurate localization, and accurate diagnosis of HCC, especially small and atypical HCCs that are not visible in conventional ultrasound [84].

In CEUS, contrast agents are composed of tiny bubbles that are encapsulated in a shell. This characteristic makes them unique in that they interact with the imaging process in such a way that they oscillate at low-intensity ultrasound fields and destroy at high-intensity ultrasound fields. Currently, available ultrasound contrast agents are available in two categories: pure blood pool contrast agents, such as Lumason (Bracco Diagnostics, Monroe Township, NJ, USA) and Definity (Lantheus Medical Imaging, Billerica, MA, USA), as well as combined blood pools and Kupffer cell contrast agents, such as Sonazid (GE Healthcare, Oslo, Norway) [85]. CEUS has unique advantages over CT and MR in that ultrasound contrast agents are not excreted by the kidneys and can therefore be safely injected in patients with renal failure or allergies to iodine or gadolinium [86,87]. CEUS also has a favorable safety profile, as shown in a retrospective analysis of 23,188 investigations. Based on a retrospective analysis of 23,188 investigations, there were no deaths reported and the reported serious adverse event rate was 0.0086% [88]. Due to the simplicity of its operation, the lack of radiation, the lack of side effects, and the high diagnostic specificity, CEUS is a suitable method for the routine monitoring of patients with small nodular cirrhosis [89,90].

#### 4.2.2. Computed Tomography

It is common for HGDN to appear as slightly hypo- or iso-dense nodules in an unenhanced CT scan with a more uniform density and poorly defined boundary [78]. A high percentage of borderline lesions are detected during the hepatic arterial, portal venous, and delayed phases of CT imaging with low-low-low, similar-low-low, or similar-similar-low CT attenuation patterns.

With an increasing level of malignancy, the portal area, including the normal portal vein (blood supply from the portal vein) and hepatic artery (blood supply from the nodule through the normal hepatic artery), also decreases. Meanwhile, that of the abnormal artery (supply to the nodule via newly formed abnormal arteries) gradually increases [85]. Although DNs have a variable and irregular blood supply, their vascular supply overlaps significantly with LGDN, HGDN, and eHCC, which have many similar imaging features [91,92]. Thus, it remains difficult to accurately describe DN and eHCC based on imaging [17,69].

While CT is a useful tool, there are some disadvantages, such as the fact that it reveals DN in cirrhosis with variable sensitivity, and that the detection rate is usually low [93,94].

#### 4.2.3. Magnetic Resonance Imaging

Conventional CT or MRI is less sensitive than MRI with hepatobiliary contrast agents for the detection of eHCC and HGDN. A combination of iron oxy-magnetic resonance imaging and CTAP or CTHA can be a viable alternative to these imaging techniques, which are both highly sensitive and have high specificity for the diagnosis of HCC [10]. In comparison with CT, MRI offers the advantages of non-radiation, high-contrast resolution, multiparameter imaging, and multidirectional imaging, among others [95]. There are several common MRI techniques, including T1WI imaging, T2WI imaging, diffusion-weighted imaging (DWI), magnetic susceptibility-weighted imaging (SWI), and so on.

For the evaluation of cirrhosis-associated nodules, there are two main types of contrast agents available, including gadolinium-based extracellular contrast agents and hepatobiliary contrast agents [96,97,98,99,100].

##### Gadolinium-Based Extracellular Contrast Agents

Extracellular contrast agents include gadoterate meglumine, gadobutrol, gadopentetate dimeglumine and so on [101]. Extracellular paramagnetic contrast agents, such as gadolinium chelates with low molecular weight, shorten T1 time and provide information about tissue vasculature. Gadolinium chelates are a subclass of drugs consisting of both extracellular and hepatocellular components [102]. These compounds, like other gadolinium chelates, can be applied to pictures taken with appropriate delay to evaluate the vascular distribution of lesions and hepatocyte function. Imaging during the arterial phase of extracellular enhancement is necessary for a thorough evaluation of the liver, especially in the late hepatic arterial phase when the portal vein is only faintly enhanced [103]. Patients with renal failure who receive low-dose gadolinium chelates for magnetic resonance imaging are generally safe, but if high doses (more than 220 mmol) are administered by arterial routes, nephrotoxicity rates can reach 40% [104,105,106]. Therefore, it is not recommended to use high doses of gadolinium chelates in clinical settings.

##### Hepatobiliary Contrast Agents

A hepatobiliary contrast agent consists of a paramagnetic compound that is absorbed by and excreted by normally functioning hepatocytes. There is an increase in signal intensity in the liver, bile ducts, and some hepatocellular lesions when these agents are used in T1-weighted imaging [101]. Gadoxetate disodium (Gd-EOB-DTPA) and gadobenate dimeglumine (Gd-BOPTA) are the primary hepatobiliary contrast agents for hepatobiliary applications [107]. It is recommended to use Gd-EOB-DTPA-enhanced MRI for the diagnosis of small HCC measuring up to 2 cm due to its higher sensitivity and diagnostic accuracy compared to multidetector computed tomography (MDCT) [108,109].

Imaging during the arterial phase of extracellular enhancement is necessary for regenerative nodules of the liver, especially in the late hepatic arterial phase when the portal vein is only faintly enhanced. Most DNs are shown as slightly hyperintense in T1WI and DWI and isointense or hypointense in T2WI, with no significant enhancement in either the arterial or delayed phases [107]. Iron is known to accumulate in LGDN and some cases of HGDN during the early phases of hepatocarcinogenesis, and these nodules appear to have a moderate-to-low signal intensity in T2-weighted fast spin-echoes and T2*-weighted GRE images [69] (Figure 2A). DN, particularly HGDN, may have a higher intracellular fat content than background liver does, resulting in hyperintensities in T1WI and a loss of signal in antiphase images. LGDN indicates a low level of signal intensity compared with the adjacent liver, whereas HGDN indicates a slightly higher level of signal intensity [107]. When liver-specific contrast agents are administered, particularly LGDN, the hepatobiliary phase tends to be hyperintense or isointense, indicating that hepatocyte function has not been significantly disrupted [110,111,112] (Figure 2B). Current MRI sequences show that, whereas HGDNs are difficult to detect from well-differentiated HCC, LGDNs are typically difficult to separate from regenerative nodules [19] (Figure 2C).

By using DWI, it is possible to gain insight into the cellular structure on the micron level as well as to determine the cellular density of hepatocellular nodules [113,114]. Hyperintensities in DWI and T2WI are more common in atypical HCC than in DN and are the strongest specificity for distinguishing eHCC from HGDN [115].

There is evidence that OATP is responsible for the uptake of two gadolinium-based contrast agents (gadolinium disodium and gadobenate glucosamine) into hepatocytes. As hepatocarcinogenesis progresses, OATP expression decreases; higher levels are found in cirrhotic nodules and LGDN, while lower levels are found in many HGDN, eHCC, and progressive HCC [54]. In the event of decreased OATP8 expression during hepatocarcinogenesis, the enhancement ratio of the hepatobiliary phase images will be reduced as a result of a decrease in Gd-EOB-DTPA uptake. Thus, hypointensity in the hepatobiliary phase is a strong predictor of precancer or malignancy, and its presence is more likely to be associated with HGDN or eHCC. In the diagnosis of liver cirrhosis and HCC, Gd-EOB-DTPA-enhanced MRI has a high level of sensitivity.

A comparison of MRI to CT indicates that MRI has a higher sensitivity for lesion detection than multidetector CT, yields a better soft tissue contrast, provides information on more tissue properties, and provides higher-contrast sensitivity. In addition, more contrast agents are available for MRI [95,116,117]. Therefore, hepatocellular nodules associated with cirrhosis are now commonly evaluated using magnetic resonance imaging (MRI).

## 5. Conclusions

As indicated by the concept of precancerous lesions of HCC, there appears to be a continuous histological continuum between LGDN and HGDN [9], and cirrhosis is a factor contributing to HGDN and HCC. Blood supply lesions are absent in DN and most eHCC cases. In terms of microscopical appearance, eHCC is characterized by the presence of small heterogeneous cells with increased density, and it is well-differentiated by an increased proportion of nuclei [56]. As opposed to DN, the sparse reticulin meshwork and interstitial invasion (tumor invasion of the fibrous septum or portal bundle) help to identify eHCC [118]. Hepatocarcinogenesis is characterized by three key changes: an increase in arterial blood flow, a decrease in portal venous blood flow, and a decrease in OATP expression. The use of CT and MRI multiphasic imaging, in conjunction with extracellular agents, can be used to diagnose HCC, based primarily on vascular assessment. Enhancement phases that are critical to the process include the late arterial, portal venous, and delayed phases [69]. MRI in conjunction with Gd-EOB-DTPA should be considered a first-line diagnostic strategy for liver nodules in patients with cirrhosis, as it provides information regarding vascular patterns and hepatocyte function [112]. Morphology should be the predominant determining factor in the diagnosis of HCC. AFP, DCP, p53, PCNA, EZ2H, as well as other tissue markers proposed by some researchers, have a certain role in the diagnosis of HCC but are of limited use in suggesting the presence of precancerous lesions, especially in the differentiation of precancerous lesions from HCC. The use of immunohistochemistry tests, such as GS and HSP70, has significantly contributed to the distinction between HCC and DN in recent years.

Currently, pathological diagnosis is still the gold standard for the diagnosis of precancerous lesions in liver cancer, but its clinical use is limited by its invasive operation and the difficulty of obtaining specimens when the nodes are small. Therefore, for most patients, the early diagnosis of HCC precancerous lesions relies more on imaging and laboratory tests. It has been reported that GPC3, GS, and HSP70 are clinically significant for differentiating precancerous lesions from benign nodules and eHCC, and more immunohistochemical indexes have been identified to improve the sensitivity and specificity of diagnosis. Among them, hepatobiliary contrast combined with MRI has a higher sensitivity for the detection of precancerous lesions. On this basis, early markers that can diagnose precancerous lesions of liver cancer should be further investigated, new markers with better specificity and sensitivity should be explored, and attempts should be made to combine multiple markers for diagnosis to compensate for the chance and limitations of single tests.

In recent years, the treatment concept of liver cancer has changed to early detection, early prevention, and early treatment, so the diagnosis of precancerous liver cancer plays a crucial role in the prognosis of liver cancer. In terms of imaging, we should focus on the exploration of more specific contrast agents and find the contrast agents that better fit the characteristics of precancerous nodules according to the nature of liver cancer, to distinguish precancerous lesions from liver cancer and benign nodules more clearly.

## Figures and Tables

**Figure 1 diagnostics-13-01211-f001:**
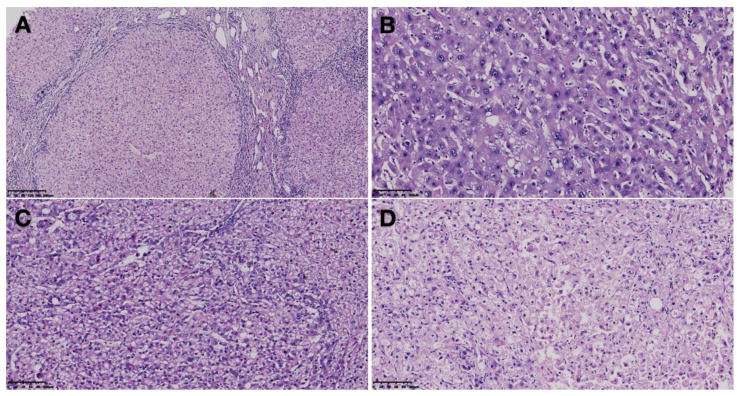
Histological characterization of hepatic cirrhosis (**A**), low-grade dysplastic nodule (LGDN) (**B**), high-grade dysplastic nodule (HGDN) (**C**), and early hepatocellular carcinoma (eHCC) (**D**). In cirrhosis, the normal lobular structure is destroyed and replaced by pseudolobules. The arrangement of hepatocytes in the pseudolobules is disturbed, accompanied by degeneration, necrosis, and regeneration of hepatocytes, and the central vein is often absent (**A**); LGDN is mainly characterized by macrocell dysplasia, with slightly increased density, portal structure, and no isolated arterioles of the pseudo glandular duct (**B**); HGDN is dominated by small cells with moderate to severe dysplasia, accompanied by structural dysplasia, and the cell arrangement density was significantly increased compared with the surrounding liver tissue (**C**); eHCC cell atypia is obvious, the cancer cells are different in size and shape, and the cancer cells are arranged in nests, with more of an interstitium (**D**). Note: Hematoxylin-esion staining, Magnification ×100 (**A**) and ×200 (**B**–**D**).

**Figure 2 diagnostics-13-01211-f002:**
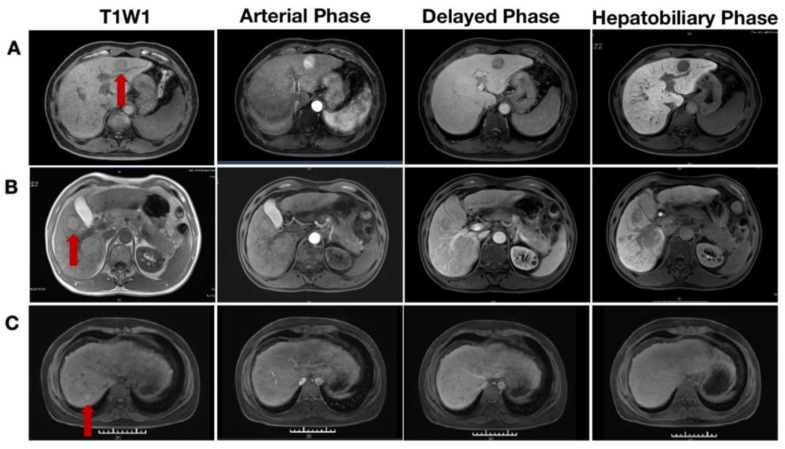
(**A**) Early hepatocellular carcinoma (eHCC); (**B**) dysplastic nodule (DN); (**C**) hepatic cirrhosis regenerative nodule (RN). In a gadolinium ethoxylate disodium-enhanced MRI, eHCC only showed hyperintensity in the arterial phase, while in the T1W1 plain scan, the delayed phase and hepatobiliary phase had slight hypointensity or hypointensity to the adjacent liver (**A**); DN appeared to have slight hyperintensity on a T1W1 plain scan in the arterial phase, showing isointensity to the adjacent liver in the delayed phase, and showing hypointensity to the adjacent liver in the hepatobiliary phase (**B**); regenerative nodules (RNs) were shown to have slight hyperintensity on a T1W1 plain scan in the arterial phase, appearing to have isointensity or slight hypointensity to the adjacent liver in the delayed phase, and having isointensity to the adjacent liver in the hepatobiliary phase (**C**).

## Data Availability

Not applicable.

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
