# Peer review of "Current Concepts of Precancerous Lesions of Hepatocellular Carcinoma: Recent Progress in Diagnosis"

_diagnostics, 2023, doi:10.3390/diagnostics13071211_

Round 1

Reviewer 1 Report

Dear Authors

I would like to thank you for the opportunity of reviewing this interesting paper that is focused on a very challenging topic also in daily clinical practice. Currently, pathological diagnosis is still the gold standard for diagnosing pre-cancerous lesions in liver cancer. Still, its clinical use is limited by the invasive operation and the difficulty of obtaining specimens when the nodes are small. Therefore, for most patients, the early diagnosis of HCC precancerous lesions relies more on imaging and laboratory tests. Therefore, papers that explore in depth imaging perspectives for HCC and its pre-cancerous lesions diagnosis could surely be of interest for this prestigious journal. Moreover, this paper expresses the aim of finding objective and practical conclusions from the many studies that have been conducted in recent years to provide results that can be used in clinical practice. I think that this paper can result endearing to many readers since an effective and reliable diagnostic tool for detecting HCC and precancerous lesions is a fundamental step for patients’ cares, therefore this topic results interesting both for radiologist and clinicians. 

This paper is pleasurable to read, although it suffers from some limitations that Authors can easily adjust in order to slightly improve their article making it more eligible for this important Journal. In particular, Authors must improve some section of the paper, adding information and including some other important references about this topic that, in my opinion, should be cited or discussed to offer a more exhaustive review of this important topic to the readers. 

First of all, although language used is appropriate, I (I am not a native English speaker) recommend to Authors to obtain a certified native speaker with proficiencies in the scientific-medical field to properly complete this paper, because several sentences are not completely fluent. Moreover, I recommend making a further revision of the title to fix some typing errors (for example, change “progression” to “progress”).

Secondly, it seems that the last Author’s name is missing, please correct.

In addition, Authors did not correctly reported keywords from MeSH Browser. This is important, in my personal opinion, in order to increase the traceability of this paper (and consequently the possibility of the Journal to be cited by Readers and Stakeholders). I suggest the check of all KW and use only those that are present in MeSH Browser.

In the Introduction section, lines 27-29: this paper is about HCC, thus do not mention “cholangiocarcinoma”, unless you talk about it, even more so if the next sentence is about the pre-cancerous lesions of HCC. 

Please be clearer regarding this aspect: “Due to the challenging early diagnosis of HCC, the prognosis of most HCC patients is disappointing, with a 5-year survival rate of ≤18”. Only approximately 20% of HCCs are diagnosed in the very early and early stages when treatments, such as liver transplantation, ablation and surgical resection, could guarantee a high 5-year survival rate. On the contrary, the majority of HCC patients are in the intermediate and/or advanced tumoural stages at presentation, and are therefore unsuitable for these treatments. They require transarterial chemoembolisation (TACE), radioembolisation, or systemic therapies, which are considered to be effective treatments but which yield a lower overall survival rate than the treatments mentioned above [Hepatol Int. 2019 Mar;13(2):125-137. doi: 10.1007/s12072-018-9919-1].

Lines 44-55 are not necessary in the Introduction, please move them in the section regarding molecular and genetic mechanism of hepatocarcinogenesis. 

Through the entire text there are some errors regarding the references [Error! Reference source not found.A] Please check all of them.

Imaging section need major revisions since many concepts are wrongly expressed and misplaced. First of all, I suggest to briefly introduce the concept that radiological features of HCC are often enough to make a correct diagnosis and biopsies are performed only when cross-sectional imaging shows atypical features, citing the following references [Radiol Med. 2022 Feb;127(2):129-144. doi: 10.1007/s11547-022-01449-w] [J Clin Med. 2022 Jul 28;11(15):4399. doi: 10.3390/jcm11154399].

Moreover, the molecular and histological characteristics of HCC subtypes and precursors often find a correspondence in their imaging features, thus introduce the parallelism between histological changes and imaging features at the beginning of this section (first the loss of AOTP transporters= hypointensity in HB-phase, then neovascularization= wash-in in arterial phase) [Histol Histopathol. 2022 Dec;37(12):1151-1165. doi: 10.14670/HH-18-487]. Please cite the aforementioned reference and expand this topic.

Similarly, the section “Computed tomography” need to talk about COMPUTED TOMOGRAPHY no CT during arterial portography (CTAP) which is now rarely performed. Please check the available literature and correct this paragraph.[ Oncology. 2007;72 Suppl 1:83-91. doi: 10.1159/000111712]

In addition, in the section “Magnetic resonance imaging” please discuss both MRI with extra-cellular contrast media and MRI with hepatobiliary contrast media. 

Finally, please check the figure. I don’t believe that B is a dysplastic nodule since is “slightly” hyperintense in arterial phase (which is, by definition, a characteristic features of overt HCC).

Best Regards,

Author Response

Response to Reviewer 1 Comments
Thank you for your precious time and kind comments on this research and also thanks for providing this valuable opportunity to revise our manuscript (Manuscript ID diagnostics-2248047). We extremely cherish this opportunity to revise and have made a lot of revisions compared with last version of manuscript followed the suggestion and comments of you and other reviewers. In response to your concerns and questions, we made the following responds point by point. We have carefully considered the suggestion of reviewer and made some changes. The point-by-point responses to the comments are listed below this letter, and all amendments are highlighted in red in the revised manuscript. (Revised manuscript enclosed at the end of the response)

Point 1: I recommend making a further revision of the title to fix some typing errors (for example, change “progression” to “progress”).

Response 1: Thank you for your valuable comments. According to your comments, we have further modified the title and changed “progress” to “progress”. Please check it.

Point 2: It seems that the last Author’s name is missing, please correct.

Response 2: We apologize for some mistakes made during the editing process and have now filled
out the names of all the authors. Please check it.

Point 3: I suggest the check of all KW and use only those that are present in MeSH Browser.

Response 3: We sincerely appreciate the valuable comments. We searched the MeSH Browser for relevant KW and adjusted the KW as follows: Replace "precancerous lesions" with "precancerous conditions", "pathological examination" with "pathology", "Imaging examination" with "Diagnostic imaging". "Dysplastic nodules" was removed because the relevant term could not be found in the MeSH Browser. Please check it.

Point 4: In the Introduction section, lines 27-29: this paper is about HCC, thus do not mention “cholangiocarcinoma”, unless you talk about it, even more so if the next sentence is about the precancerous lesions of HCC.

Response 4: We are very grateful to reviewer for reviewing the paper so carefully. Given that there is no subsequent mention of cholangiocarcinoma in this article, we have followed your suggestion and removed that sentence. Please check it.

Point 5: Please be clearer regarding this aspect: “Due to the challenging early diagnosis of HCC, the prognosis of most HCC patients is disappointing, with a 5-year survival rate of ≤18”. Only approximately 20% of HCCs are diagnosed in the very early and early stages when treatments, such as liver transplantation, ablation and surgical resection, could guarantee a high 5-year survival rate. On the contrary, the majority of HCC patients are in the intermediate and/or advanced tumoural
stages at presentation, and are therefore unsuitable for these treatments. They require transarterial chemoembolisation (TACE), radioembolisation, or systemic therapies, which are considered to be effective treatments but which yield a lower overall survival rate than the treatments mentioned above [Hepatol Int. 2019 Mar;13(2):125-137. doi: 10.1007/s12072-018-9919-1].

Response 5: Thank you for your valuable advice. We have reviewed the relevant literature, and on this basis, combined with your valuable suggestions, we have deleted this sentence and restated the prognosis of liver cancer treatment (line 30-35). Please check it.

Point 6: Lines 44-55 are not necessary in the Introduction, please move them in the section regarding molecular and genetic mechanism of hepatocarcinogenesis.

Response 6: Thank you for your valuable comments. We reviewed this introduction. Following your suggestion, we added a new section "molecular and genetic mechanism of hepatocarcinogenesis", and moved the relevant contents in the introduction to this section and made some adjustments (line 108-122). Please check it.

Point 7: Through the entire text there are some errors regarding the references [Error! Reference source not found. A] Please check all of them.

Response 7: Once again, we deeply apologize for the error in the editing process. We found that the wrong references were all derived from the picture citations, and we have corrected them (line 47, 83, 92, 102, 345, 351, 353). Please check it.

Point 8: I suggest to briefly introduce the concept that radiological features of HCC are often enough to make a correct diagnosis and biopsies are performed only when cross-sectional imaging shows atypical features, citing the following references [Radiol Med. 2022 Feb;127(2):129-144. doi: 10.1007/s11547-022-01449-w] [J Clin Med. 2022 Jul 28;11(15):4399. doi: 10.3390/jcm11154399].

Response 8: We would like to express our heartfelt thanks to you for your very valuable advice. Based on your suggestions, we briefly describe the typical imaging features of HCC at the beginning of the "imagine examination" section and cited relevant literature (line 215-222). Please check it.

Point 9: Moreover, the molecular and histological characteristics of HCC subtypes and precursors often find a correspondence in their imaging features, thus introduce the parallelism between histological changes and imaging features at the beginning of this section (first the loss of AOTP transporters= hypointensity in HB-phase, then neovascularization= wash-in in arterial phase) [Histol
Histopathol. 2022 Dec;37(12):1151-1165. doi: 10.14670/HH-18-487].

Response 9: Thank you again for your valuable suggestions which lead a promising direction for this manuscript. According to your suggestion, we searched and summarized the relevant literature, obtained a general understanding of the imaging findings, and briefly explained the connection
between histology and imaging in HCC (line 223-232). Please check it.

Point 10: “Computed tomography” need to talk about COMPUTED TOMOGRAPHY no CT during arterial portography (CTAP) which is now rarely performed. Please check the available literature and correct this paragraph. [ Oncology. 2007;72 Suppl 1:83-91. doi: 10.1159/000111712].

Response 10: We sincerely appreciate the valuable comments. We revised this part to remove the contents related to CTAP and CTHA, and focused on the contents related to COMPTED TOMOGRAPHY (line 289-302). Please check it.

Point 11: In the section “Magnetic resonance imaging” please discuss both MRI with extra-cellular contrast media and MRI with hepatobiliary contrast media.

Response 11: Thank you for your valuable comments. According to your suggestion and combined with our literature summary, we have removed the discussion of MRI with superparamagnetic iron oxide contrast agents in the original manuscript. It was changed to a discussion of MRI with Gadolinium-based extracellular contrast agents and Hepatobiliary contrast agents (line 312-337). Please check it.

Point 12: Please check the figure. I don’t believe that B is a dysplastic nodule since is “slightly” hyperintense in arterial phase (which is, by definition, a characteristic features of overt HCC).

Response 12: Thank you very much for your questions about the Figure B, article in ‘MR with Gd- EOB-DTPA in assessment of liver nodules in cirrhotic patients’ showed that depending on iron or fat concentration, HGDNs can differently appear on pre-contrast sequences. In our figure, B is the
image of HGDN, because there is no difference between the image of LGDN and RN, we chose the image of HGDN. The arterial phase showed a slightly elevated signal because there was a slightly higher signal in flat phase. If the subtraction image from the arterial phase is subtracted from the
plain scan, it will not be high signal. [Inchingolo R, Faletti R, Grazioli L, et al. MR with Gd-EOB-DTPA in assessment of liver nodules in cirrhotic patients. World J Hepatol. 2018 Jul 27;10(7):462-473. doi: 10.4254/wjh.v10.i7.462. PMID: 30079132; PMCID: PMC6068846.]

Reviewer 2 Report

Diagnostics-2248047 “Current concepts of precancerous lesions of the hepatocellular carcinoma: recent progression in diagnosis” well explained HCC and precancerous lesions with pathologic and radiologic respects.

But I suggest minor problems.

1. I would like you to have a professional English proofreading because some sentences seemed to be inappropriate.

2. I noticed some reference errors in the paper that need to be corrected.

Author Response

Response to Reviewer 2 Comments

Thank you for your precious time and kind comments on this research and also thanks for providing this valuable opportunity to revise our manuscript (Manuscript ID diagnostics-2248047). We extremely cherish this opportunity to revise and have made a lot of revisions compared with last version of manuscript followed the suggestion and comments of you and other reviewers. In response to your concerns and questions, we made the following responds point by point. We have carefully considered the suggestion of reviewer and made some changes. The point-by-point responses to the comments are listed below this letter, and all amendments are highlighted in red in the revised manuscript. (Revised manuscript enclosed at the end of the response)

Point 1: I would like you to have a professional English proofreading because some sentences seemed to be inappropriate.

Response 1: We sincerely appreciate the valuable comments. Our manuscript has been revised under the guidance of several experts. We have consulted Elsevier Language Editing Services (Order reference: ASLESTD0189347) (a language editing certificate attached at the end of the letter) and invited experts for English writing to help us with editing our English manuscript. There may still be some language problems in our manuscript, but due to the short time of this revision, the language polishing of professional institutions cannot be returned within the specified time, so we have made preliminary revision temporarily. If the manuscript is accepted later, we will make further revision in the language polishing company of your magazine before publication.

Point 2: I noticed some reference errors in the paper that need to be corrected.

Response 2: We deeply apologize for the error in the editing process. We found that the wrong references were all derived from the picture citations, and we have corrected them (line 47, 83, 92, 102, 345, 351, 353). Please check it.

Round 2

Reviewer 1 Report

Authors addressed raised points appropriately.